# Engineering the Quaternary Hydrotalcite-Derived Ce-Promoted Ni-Based Catalysts for Enhanced Low-Temperature CO_2_ Hydrogenation into Methane

**DOI:** 10.3390/ma16134642

**Published:** 2023-06-27

**Authors:** Yuxin Peng, Xin Xiao, Lei Song, Ning Wang, Wei Chu

**Affiliations:** 1School of Chemical Engineering, Sichuan University, Chengdu 610065, China; pyx18908007922@163.com (Y.P.); songlei@scu.edu.cn (L.S.); 2College of Carbon Neutrality Future Technology, Sichuan University, Chengdu 610106, China; 3National Engineering Research Centre for Flue Gas Desulfurization, Chengdu 610065, China; 4College of Environmental and Energy Engineering, Beijing University of Technology, Beijing 100124, China; ning.wang.1@bjut.edu.cn

**Keywords:** Ce dopants, Ni-based catalyst, CO_2_ methanation, lower temperature performance, quaternary hydrotalcites

## Abstract

Ce-promoted NiMgAl mixed-oxide (NiCex-C, x = 0, 1, 5, 10) catalysts were prepared from the quaternary hydrotalcite precursors for CO_2_ hydrogenation to methane. By engineering the Ce contents, NiCe5-C showed its prior catalytic performance in low-temperature CO_2_ hydrogenation, being about three times higher than that of the Ce-free NiCe0-C catalyst (turnover frequency of NiCe5-C and NiCe0-C: 11.9 h^−1^ vs. 3.9 h^−1^ @ 225 °C). With extensive characterization, it was found that Ce dopants promoted the reduction of NiO by adjusting the interaction between Ni and Mg(Ce)AlOx support. The highest ratio of surface Ni^0^/(Ni^2+^ + Ni^0^) was obtained over NiCe5-C. Meanwhile, the surface basicity was tailored with Ce dopants. The strongest medium-strength basicity and highest capacity of CO_2_ adsorption was achieved on NiCe5-C with 5 wt.% Ce content. The TOF tests indicated a good correlation with medium-strength basicity over the NiCex-C samples. The results showed that the high medium-strength and Ce-promoted surface Ni^0^ species endows the enhanced low-temperature catalytic performance in CO_2_ hydrogenation to methane.

## 1. Introduction

Recently, CO_2_ reduction and recycling has become one of the most researched hotspots in neutral carbon economics due to the serious impact on global climate change led by growing CO_2_ emissions [1,2]. To mitigate the increasing anthropogenic CO_2_, methods like CO_2_ capture, utilization, and storage (CCUS) had been engaged [3]. Among these methods, the conversion of CO_2_ with “green hydrogen” (hydrogen originates from sustainable energy, like wind, solar, etc.) into synthetic natural gas (SNG) has been considered as one of the most promising and practical approaches for CO_2_ utilization [4,5]. SNG, as a high value-added fuel product, plays a vital role as a raw material in the synthesis of syngas and many chemical products [6]. CO_2_ hydrogenation to the SNG reaction, which was also called the Sabatier reaction, displayed a great potential in the “Power to Gas” process [7]. However, this reaction went through an eight-electron process for CO_2_ reduction into methane, which suffered significant kinetic limitations [8]. To solve this puzzle, a catalytic material that could achieve high reaction rates was required. Ni-based catalysts with high reaction activity and low-cost aroused extensive interest of researchers [9]. Meanwhile, in practical application, Ni-based catalysts always suffered from inferior activity at low temperatures [10]. Compared to the CO_2_ methanation reaction in a higher temperature, the low-temperature CO_2_ methanation showed advantages in two different aspects: (I) The competitive reaction, like the reverse water gas shift (RWGS) reaction, could be effectively reduced at low temperatures, and the catalysts obtained a high CH_4_ selectivity. (II) The sintering and carbon deposition problems of the catalyst usually occurred at high temperatures; the low temperature condition was conducive to the catalyst’s stability [11,12,13]. Therefore, improving the low temperature activity of Ni-based catalysts is a current research hotspot [14].

The catalytic activity of Ni-based catalysts is usually related to the basic nature of the catalyst carrier and the preparation method [15]. Compared with Ni/γ-Al_2_O_3_ catalysts prepared by impregnation, Ni-Al hydrotalcite catalysts prepared by co-precipitation could contain a higher content of well-dispersed Ni nanoparticles with more reducible Ni species [16]. Hydrotalcite (HT) was a kind of natural or synthetic ordered material which consisted of positively charged two-dimensional sheets of mixed hydroxides and charge-compensating anions placed between the layers [17]. They were expressed as [M^2+^_1−x_M^3+^_x_(OH)_2_] (A^n−^)_x/n_ · mH_2_O (M^2+^ and M^3+^ were divalent and trivalent metals, respectively; x was the mole ratio of M^3+^/(M^2+^ + M^3+^), A^n−^ was the interlayer anion) [7]. The unique supramolecular structure provided great potential to disperse and tune active sites at the atomic scale [18]. Introducing a proportion of alkaline elements into the HT precursor could obtain a tunable alkaline site structure and promote CO_2_ adsorption, which would benefit the CO_2_-involving reaction. For instance, the addition of the Mg element could improve the catalyst’s basic properties, result in the increase of the CO_2_ adsorption capacity [19]. Moreover, small-size and heat-stable metal nanoparticles were highly dispersed on the calcined HT precursor surface after reduction; thus, the catalyst stability and reducibility would get enhanced [10]. For example, Guo et al. [20] synthesized that the hydrotalcite-derived NiMgAl catalyst exhibited excellent catalytic activity with a CO_2_ conversion of 91.8% at 250 °C. Therefore, it was a promising approach by fabricating the alkaline-assisted hydrotalcite-derived materials to obtain the efficient low-temperature catalyst for CO_2_ methanation.

However, traditional NiMgAl catalysts derived from hydrotalcite also suffered from strong metal-support interactions [21]; thus, a long-time H_2_ reduction process was needed to enhance reducibility. To overcome this drawback, quite a few reports were found to introduce a second metal (such as Mn, La, and Y) to replace the partial Mg element [22,23], thereby regulating the metal-support interaction. As reported by our group [11], the doping of the Mn element could efficiently regulate the Ni and Mg(Mn)AlOx interaction, surface content of Ni^0^ species, and basic property. Dominik Wierzbicki et al. [24] supposed La could soften the interaction between Ni and the HT matrix and lead to the increase of Ni-species reducibility [25]. Sun et al. found the incorporation of the 0.4 wt.% Y strongly decreased the metallic nickel particle size and increased the medium-strength basic sites. 

Besides the Mn, La, Y species, CeO_2_, as a rare earth oxide, had also been a very attractive promoter for CO_2_ methanation due to its extraordinary ability to enhance metal dispersion, as well as the thermal stability of the support [26,27]. Moreover, the rare earth metal element Ce owns redox properties [28]. During the reaction, Ce metal ions are reduced to produce oxygen vacancies, which are the sites of dissociation and activation of CO_2_ [28,29]. Radosław Debek et al. [30] reported CeO_2_ had a promoting effect on increasing the surface basicity of the catalyst, which was attributed to its high mobile oxygen capacity and redox activity. The process of reducing CO_2_ to CO was the rate-determining step of CO_2_ methanation. In addition, the oxygen vacancy provided by the CeO_2_ could create an additional driving force for this process in the reducing atmosphere [31]. Zhang et al. [32] considered that Ce^3+^ cations located in the AlCeO_3_ solid solution could greatly promote the adsorption and activation of CO_2_ and facilitate the formation of the intermediate; therefore, the CO_2_ conversion was significantly accelerated at 200 °C. In the dry reforming of methane (DRM) reaction, Radosław Debek et al. [17,33] incorporated Ce species into the NiMgAl catalysts, which was found to promote the nickel species’ reducibility and introduce new strong oxygen species (low coordinated) and more medium-strength basic sites (Lewis acid-base pairs). 

Furthermore, a recent literature compared different Ni content (10.3, 16.2, 27.3, 36.8, 42.5 wt.%) on the performance of hydrotalcite-derived catalysts, and the optimum Ni content was 42.5 wt.%. This result confirmed that higher amounts of Ni introduced led to smaller crystallites, better reducibility, and CO_2_ adsorption capacity of the catalysts. H_2_-TPR proved the Ni and hydrotalcite matrix interaction weakened with the increasing Ni contents, which had a positive effect on the catalytic activity [34]. However, according to our knowledge, quaternary hydrotalcite-derived Ce-containing Ni-Ce catalysts had not been widely used in CO_2_ methanation. Herein, our goal was to study the structure–performance relationships between them. We prepared a series of Ce-promoted Ni/MgAlOx catalysts through the co-precipitation method in this study. The low-temperature catalytic performance was investigated by varying Ce content from 0, 1, 5 to 10 wt.%. The optimal sample displayed superior CO_2_ methanation activity with 80% CO_2_ conversion at a low temperature of 250 °C, and the CH_4_ selectivity was close to 100%. Meanwhile, the CO_2_ conversion (91.7%) was not deactivated upon 80 h of operation at 300 °C. Extensive characterization methods (XRD, BET, ICP SEM, TEM, XPS, H_2_-TPR, CO_2_-TPD) were used to deeply analyze the promoting effect of Ce on influencing the catalyst structure, morphology, surface properties, metal-support interaction, and performances during CO_2_ methanation. 

## 2. Experimental

### 2.1. Catalyst Synthesis

All of the hydrotalcite precursors were prepared by the co-precipitation method. Ni content was kept at 40 wt.% and the Mg/Al molar ratio was fixed at 1 in all samples, while the loading of Ce varied from 0, 1, 5 to 10 wt.%. Firstly, the mixture of Ni(NO_3_)_2_·6H_2_O, Mg(NO_3_)_2_·6H_2_O, Al(NO_3_)_3_·9H_2_O, and Ce(NO_3_)_3_·6H_2_O was dissolved in deionized water to form a solution at 0.4 M. Then, the mixed nitrate solution was added dropwise into a flask containing 280 mL sodium carbonate solution (0.25 M) at 60 °C under vigorous stirring, while at the same time, keeping the slurry pH at 9.5–10 by adding the sodium hydroxide solution (1 M). After co-precipitation, the slurry was vigorously stirred at 60 °C for 1 h and then aged for 18 h at 60 °C. The solid product was obtained by filtration, washed with deionized water three times, dried at 60 °C overnight, and named as NiCe0-HT (without Ce) or NiCex-HT (x = 1, 5, 10). Finally, the obtained hydrotalcite precursors were calcined at 500 °C for 4 h and labeled as NiCex-C (x = 0, 1, 5, 10).

### 2.2. Catalytic Experiments

The CO_2_ methanation reaction was performed in a miniature fixed-bed reactor (Xingtianyu, Chengdu, China) at atmospheric pressure. A total of 200 mg of the catalyst (40–60 mesh) mixed with 500 mg of quartz sand were placed in the quartz tube (inner diameter of 6mm and a length of 40 cm). Prior to the catalyst evaluation, the catalysts were reduced under 30 mL/min H_2_ at 700 °C for 2 h. After reduction, the catalyst bed was cooled down in the N_2_ atmosphere to 180 °C; then, the mixture of 40 mL/min H_2_ and 10 mL/min CO_2_ (H_2_:CO_2_ = 4:1, GHSV = 15,000 mL/g/h) was introduced into the reactor. The catalytic test was carried out in the temperature range of 200–350 °C at a temperature interval of 25 °C. The gas products were analyzed by an online SP-7890 gas chromatograph (Lunan Ruihong, Zaozhuang, China) equipped with a thermal conductivity detector (TCD, Xian quan, Tianjing, China) (fitted with a TDX01 column). The conversions of CO_2_ (*X*_CO_2_)_ and the selectivity of CH_4_ (*S*_CH_4__) were calculated by the following equations [35]:(1)$$\;X_{{\mathrm{CO}}_{2}}=\left(\frac{F_{{\mathrm{CO}}_{2},\mathrm{in}}-F_{{\mathrm{CO}}_{2},\mathrm{out}}}{F_{{\mathrm{CO}}_{2},\mathrm{in}}}\right)\times 100\%$$
(2)$$\;S_{{\mathrm{CH}}_{4}}=\left(\frac{F_{{\mathrm{CH}}_{4},\mathrm{out}}}{F_{{\mathrm{CO}}_{2},\mathrm{in}}-F_{{\mathrm{CO}}_{2},\mathrm{out}}}\right)\times 100\%$$
where “*F*_CO_2_,in_” and “*F*_CO_2_,out_” refer to the inlet CO_2_ gas flow and the outlet CO_2_ gas flow, respectively, and “*F*_CH_4_,out_” refers to the outlet CH_4_ gas flow, mL/min. 

The equations of the CO_2_ conversion rate (*R*_CO_2__, μmolCO_2_/(g_cat_·s)) and turnover frequency (*TOF*, h^−1^) are as follows [36,37].
(3)$$\;\;\;\;\;\;R_{{\mathrm{CO}}_{2}}=\frac{F_{{\mathrm{CO}}_{2,}\mathrm{in}}\times X_{{\mathrm{CO}}_{2}}}{m\times V_{m}}$$
where V_m_ represents the gas molar volume of 22.4 L/mol under standard conditions, *X*_CO_2__ refers to the conversion of CO_2_ at 225 °C (*X*_CO_2__ < 15%), and m denotes the quantity of the catalyst (g) [11,38]
(4)$$TOF=\frac{F_{{\mathrm{CO}}_{2,}\mathrm{in}}\times \delta }{N_{\mathrm{surface}}\times V_{m}}$$
(5)$$N_{\mathrm{surface}}=N_{\mathrm{total}}\times D$$
where *δ* represents the methane yield and $N_{\mathrm{surface}}$ represents the molar amount of nickel atoms located on the catalyst surface, which is calculated by the nickel dispersion equation (D, %) based on the size of the Ni nanoparticles after reduction (TEM result).

The apparent activation energy of the catalyst was measured according to the Arrhenius equation:(6)$$\mathrm{lnk}={\mathrm{lnk}}_{0}-\frac{E_{a}}{\mathrm{RT}}$$
where k is the reaction rate, k_0_ is the pre-exponential factor, Ea is the apparent activation energy of the reaction, R is the gas reaction constant (8.314 J/(mol·K)), and T is the reaction temperature. The experimental conditions were carefully selected to obtain the catalytic data with CO_2_ conversion below 15% for the activation energy calculation, which effectively excluded the influence of internal and external diffusion [38].

### 2.3. Characterization of Catalysts

A powder X-ray diffraction (XRD) measurement was performed on a Rigaku Ultima IV (Japan) device equipped with a copper-based anode (Cu Kα radiation, λ = 0.154 nm). The instrument settings were 35 kV × 30 mA. It operated in the 2θ range from 5° to 85° with the scanning speed of 2°·min^−1^ to identify the crystal phase and morphology of the sample.

The hydrogen temperature-programmed reduction (H_2_-TPR) measurement was performed on a TP-5080 instrument equipped with a TCD detector (Xianquan, Tianjing, China) to investigate the reduction performance of the catalyst. Firstly, 50 mg of the catalyst was placed in the quartz tube, then 30 mL/min N_2_ was put into the instrument for 1 h at 200 °C to remove the physical adsorbed impurities on the catalyst’s surface. After cooling down to 50 °C under argon, 28 mL/min 10% H_2_/N_2_ mixture was introduced to reduce the catalyst, and the system was kept at 50 °C for 40 min until the baseline was stable. Then, the reactor temperature was linearly increased from 50 °C to 800 °C at a heating rate of 5 °C/min. 

The carbon dioxide temperature-programmed desorption (CO_2_-TPD) was carried out on an Auto Chem II 2920 apparatus to determine the basic sites on the catalyst surface. Firstly, 0.1 g catalyst was placed in the quartz tube, and 30 mL/min H_2_/Ar was put into the device for 10 min at room temperature to remove the residual gas in the tube. Then, the catalyst was reduced in H_2_/Ar at 700 °C for 2 h. After that, the sample was cooled down to 50 °C in the N_2_ atmosphere. After CO_2_ adsorption of the catalyst for 1 h, N_2_ was put into the system to remove the remaining CO_2_ in the gas phase and the physically adsorbed CO_2_ on the catalyst surface. Finally, the system was heated from 50 °C to 800 °C at a linear heating rate of 10 °C/min to desorb the chemisorbed CO_2_ on the catalyst surface.

The N_2_ adsorption-desorption analysis was carried out on the ASAP 2020 (Micrometrics, Norcross, GA, USA) analyzer at −196 °C. All samples were degassed at 300 °C for 6 h before the analysis to desorb contaminants and moisture. The specific surface area was characterized by an adsorption isotherm according to the multiple Brunauer-Emmett-Teller (BET) equation. The pore size distribution and average pore diameter were determined by the Barrett-Joyner-Halenda (BJH) model. Inductively coupled plasma optical emission spectrometer (ICP-OES, Agilent, Santa Clara, CA, USA) was carried out by Agilent 5110(OES) equipment to conduct elemental analysis of the catalysts.

X-ray photoelectron spectroscopy (XPS) measurements were performed over a K-Alpha spectrometer (Thermo Scientific, Waltham, MA, USA) equipped with a Monochromated Al Kα X-ray source (hυ = 1486.6 eV, 12 kV, 6 mA). The binding energy standard was C1s = 284.80 eV.

A ZEISS Sigma 30 scanning electron microscope (SEM) was used to study the morphology of the catalyst precursor. The acceleration voltage was 3 kV, and the magnification was 5 W and 10 W times. Moreover, the morphology, the particle size distribution, and the lattice spacing of the calcined and reduced catalysts were characterized by field emission transmission electron microscopy (TEM), which was performed on an FEI Tecnai G2 F20 instrument (USA) with an acceleration voltage of 200 kV.

## 3. Results and Discussion

### 3.1. Texture Characteristics of the NiCex-C Catalysts

Figure 1a showed the XRD diffraction profiles of precursors with various Ce contents. Distinct diffraction peaks were observed at 2θ~11°, ~22°, ~34°, ~38°, ~46°, ~60°, and ~61° for all the samples, which belonged to (003), (006), (012), (015), (018), (110), and (113) of the hydrotalcite characteristic structure (JCPDS-22-0700) [11]. In particular, compared with the Ce-free sample, obviously weaker and wider diffraction peaks were observed in other Ce-containing samples. These results indicated that the Ce doping could influence the crystallinity of the sample and disturb the hydrotalcite plate layer structure [39]. Figure 1b showed the XRD spectrum of samples after calcination at 500 °C for 4 h, and it could be found that the disappearance of all the diffraction peaks corresponded to hydrotalcite-like structures. The peaks at 2θ = 37.3°, 43.3°, 62.9°, and 75.5° were corresponded to (111), (200), (220), and (311) crystal planes of NiO, respectively (JCPDS-47-1049) [40]. After adding the Ce element, the intensity of these characteristic peaks in the samples decreased significantly. Meanwhile, well-crystallized CeO_2_ could be observed after calcination. The peaks at 28.5°, 33.0°, 47.4°, and 56.3° were indexed to (111), (200), (220), and (311) crystal planes of CeO_2_ (JCPDS-34–0394) [41]. In special, the diffraction peaks corresponding to MgO and Al_2_O_3_ were absent. The Al and Mg species may form as an amorphous structure or as a part of the MgAl_2_O_4_ spinel, whose diffraction peaks were overlapped by the periclase [11,42]. 

Texture properties of the NiCex-C catalysts were investigated by N_2_ adsorption-desorption analysis. As presented in Figure 2a, according to the IUPAC classification, all the samples displayed an IV N_2_ adsorption-desorption isotherm (P/P_0_ > 0.4) with H2 hysteresis loops, corresponding to the typical features of mesoporous structures [3]. The pores might originate from the stack of the 2D structure of HT, which was conducive to the adsorption and activation of active gases, promoting the mass transmission process [43]. As shown in Figure 2b, all samples featured typical mesoporous structures with narrow pore size distributions, concentrated from 2 to 10 nm. In addition, the detailed textural properties of the NiCex-C catalysts were listed in Table 1. All the catalysts derived from hydrotalcite-like compounds displayed large specific surface areas (>100 m^2^/g), which allowed for a better dispersion of surface nickel species [44]. The mean pore diameters of the catalysts were all slightly decreased after the introduction of the Ce element, mainly due to the Ce species on the external porous surface of the hydrotalcite crystallites, causing the blockage of partial smaller mesoporous [45]. On the other hand, there was no significant difference in the average pore volume among all samples.

### 3.2. Morphological Study and the Particle Size Analysis

The SEM images of the NiCe0-HT sample with the magnification of 10W and 5W were shown in Figure 3a and Figure 3c, respectively. It was found that the NiCe0-HT exhibited platelet-like crystals that were aggregated as rosettes, referring to the typical characteristic for the hydrotalcite structure [46]. However, the morphology of the Ce-loading sample (NiCe5-HT) was found unchanged (Figure 3b,d). It further indicated that the hydrotalcite-like structure could be well-generated, which was identical to the XRD results. 

TEM measurements were used to observe the state of all the calcined and reduced NiCex-C catalysts. As shown in Figure 4, the dark spots were ascribed to the Ni nanoparticles dispersed on the frame of the mixed metal oxide. For all samples, the layered structure of the hydrotalcite precursors partially collapsed after calcination. The morphology and the Ni particle dispersion were varied due to the introduction of different Ce contents. For the Ce-containing catalysts, Ni nanoparticles dispersed evenly on the catalyst surface. While large clusters of Ni particles were observed on the surface of the NiCe0-C catalyst, the dispersion of Ni nanoparticles was poor. In addition, the well-defined lattice fringes assigned to the face-centered cubic Ni (111) surface were observed in HRTEM images for all samples, with an average lattice spacing of 0.2 nm [4].

The particle-size frequency distribution and the average particle size were obtained by calculating the particle size of about 100 particles in the TEM image and plotting the frequency distribution histogram. As displayed in Figure 5, average particle size is in the range of 5–10 nm. The small size of the Ni nanoparticles is probably due to the confinement effect of the HT precursor [47], which processed that the ordered metal oxides grids would confine the growth of the Ni particles and achieve the highly dispersed Ni particles [48,49]. Meanwhile, the corresponding average particle size was found to follow the order: NiCe5-C (5.67 nm) < NiCe1-C (6.82 nm) < NiCe10-C (7.71 nm) < NiCe0-C (9.06 nm). Optimal Ce element loading (5 wt.%) led to the lowest Ni particle size. In a word, the statistics and TEM images proved that the aggregation of Ni particles in the NiCe0-C catalyst could be halted to some extent by the incorporation of the Ce element, which would play a positive effect on the uniform dispersion of the Ni nanoparticles.

### 3.3. Reducibility and Metal-Support Interaction Study

The reduction behavior of the NiCex-C (x = 0, 1, 5, 10) catalysts was evaluated by H_2_-TPR analysis. H_2_ consumption during the H_2_-TPR experiments should be caused mainly by the reduction of the NiO species. It was known that the reduction difficulty of the NiO species might be determined by the metal-support interaction with the oxide support or the dispersion conditions over the support surface [41]. As shown in Figure 6, three types of reduction peaks could be observed. The low-temperature reduction peak (340~360 °C) of the NiO species can be ascribed to the weakly interacting Ni species with the support [50]. It could be found that the peak intensity (340~360 °C) gradually became weaker with the increasing of Ce contents. The peak temperature decreased to 346 °C over the NiCe1-C catalyst, compared to 360 °C for undoped NiCe0-C. These results indicated that after the addition of the Ce element, this part of NiO was more easily reducible [51]. The minor shoulder peak (~510 °C) was attributed to the reduction of a small amount of Ni^2+^ species, which are weakly bound to oxygen atoms at the Ni-O-Al (or Ce) interface, commonly reported in Ni/AlCeO-x catalysts [32]. Besides, for Ce-containing catalysts, this reduction peak also corresponded to the reduction of Ce^4+^ to Ce^3+^ species [52]. The main reduction peak appeared at 700 °C for the NiCex-C catalysts, which was attributed to a stronger metal-support interaction between the NiO species and the support matrix (Mg(Ce)AlO) [26]. To obtain highly reduced samples, the reduction temperature in the activity test was chosen to be about 700 °C. The reduction temperature of NiO in the NiCe5-C catalyst especially shifted to the lowest temperature of 690 °C. Consequently, for these NiO species, the NiCe5-C catalyst displayed the best reduction behavior. However, when the Ce element was incorporated into the catalyst, the high temperature peak (~700 °C) became broader with lower intensity. This result indicated that the strong metal-support interaction was weakened by the loading of the Ce element [53]. Appropriate metal-support interactions would lead to more Ni^0^ active sites [54], thus giving preferable low-temperature catalytic activity.

### 3.4. Surface Basicity and Element Distribution Analysis

In the CO_2_ methanation process, the reaction molecule CO_2_ was not directly absorbed on metal nickel, but on surface basic sites (OH^−^ groups or alkali/alkaline-earth metal oxides), which was beneficial to activate CO_2_ [21,55,56]. Hence, CO_2_-TPD was carried out for the NiCex-C catalysts to determine the basic strength and CO_2_ adsorption capacity. As shown in Figure 7, the desorption curves of all the samples could be divided into three types of Gaussian peaks (α, β, γ peaks). The three reduction peaks occurred at 100~120, 140~170, and 250~280 °C, respectively [7,11], which correspond to weak-strength basic sites (α and β peaks) and medium-strength basic sites (γ peak). The weak-strength basic sites were attributed to the rapid formation of bicarbonate, which was due to the weak adsorption of CO_2_ by the hydroxyl group on the catalyst surface [14], while the medium basic sites were assigned to Lewis basic sites (acid-base Ce^4+^-O^2−^ and metal-O^2−^ pairs) associated with CO_2_ adsorption. These basic sites bound CO_2_ sufficiently strong for its activation and subsequent reduction [57]. Based on the literature [58], the medium basic sites played an important role in the CO_2_ methanation process. From the previous report, the surface charge would be disturbed by introducing hetero-ions into the hydrotalcite layer [59]. Thus, the incorporation of the Ce element into the nickel-based hydrotalcite obviously influenced the distribution of basic sites. Detailed information of the surface basic sites’ relative content for all reduced NiCex-C samples was displayed in Table 2. The proportion of the medium basic sites were calculated and were found to follow the order: NiCe5-C > NiCe1-C > NiCe0-C > NiCe10-C. The quantities of CO_2_ adsorbed on basic sites over reduced NiCex-C catalysts were summarized in Table 3. The highest density of medium-strength basic sites was found on the NiCe5-C sample (0.79 mmol CO_2_/g_cat_), suggesting that CO_2_ could be more easily activated and further affect the catalytic performance. The incorporation of 10 wt.% of the Ce element could lead to a slight decrease in total basicity (0.80 mmol CO_2_/g_cat_). The NiCe10-C exhibited the lowest CO_2_ adsorption capacity, and thus, the least amount of CO_2_ for the methanation. The correlation between the surface basic sites and its catalytic activity in the CO_2_ methanation will be further explored in subsequent sections [60].

The information of the surface components for the NiCex-C catalysts was further probed by XPS measurements. It was known that the position of the most intense peak was used to confirm the oxidation state of the metallic element and to obtain the information about the charge density for its cations [31]. As displayed inFigure 8a, the main band of the Ni 2p_3/2_ spectra for all the NiCex-C catalysts were deconvoluted into three peaks at ~853.8, ~856.6, and ~862.3 eV, corresponding to Ni^0^, Ni^2+^ (bulk NiO), and satellite peak of nickel species [44]. From the literature, the satellite peak was assigned to Ni^2+^ species in the Ni-O-Ce interaction interface or Ni^2+^ species in unreducible NiAl_2_O_4_ spinel [61]. These Ni species at higher binding energy (~856 and~862 eV) suggested the strong metal-support interaction [62]. Based on the peak fitting and peak area calculation, the detailed information of Ni^0^ species’ relative contents were listed in Table 4. Compared with the Ce-free sample, the Ni^0^ relative content increased. In particular, the maximum value was 40.5% for the NiCe5-C catalyst, while the minimum value was 31.5% for the NiCe0-C catalyst. It was shown that the strong interaction between Ni^2+^ and support was weakened by the Ce dopants, leading to more Ni^0^ species generated, which was consistent with the H_2_-TPR results. Rare earth oxides might act as an electron donor so that more electrons were shifted to the Ni species, resulting in an increase of d-electron density on the Ni surface, similar to the reported (Sc, Y, Ce, and Pr) promoted NiMgAl catalysts for dry reforming of methane [63]. Meanwhile, this result also indicated that the introduction of Ce to the catalysts could promote the electron transfer and reduce more Ni^2+^ to Ni^0^. While the incorporation of Ce contents was further increased (10%), the proportion of Ni^0^ shifted to the lower value (34.1%). 

On the other hand, the two sets of Ce 3d spin–orbit coupling peaks, corresponding to 3d3/2 (labeled as u) and 3d5/2 (labeled as v), respectively, could be deconvoluted into eight peaks (Figure 8b). According to the literature, the oxidation states of cerium species mainly existed in the form of Ce^3+^ and Ce^4+^ [1]. More precisely, the peaks near 882.8 eV (v), 887.8 eV (v_2_), 899.2 eV (v_3_), 901.9 eV (u), 908.1 eV(u_2_), and 917.5 eV(u_3_) could be attributed to the Ce^4+^ species in CeO_2_ [50]. However, for the NiCe1-C catalyst with lower Ce-loading, the u_3_ peak near 917.5 eV was not noticeable from XPS spectra. Additionally, the peak near 885.3 eV(v_1_) 904.9 eV (u_1_) could be ascribed to the Ce^3+^ state [64]. The Ce^3+^ state indicated the reduction of Ce^4+^ due to the high-temperature hydrogen reduction [45]. In order to keep charge balance of CeO_2_, oxygen vacancies were generated in this reduction process [50,65]. Thus, the concentration of Ce^3+^ was in direct proportion to oxygen vacancy contents, which were the main active sites in CO_2_ activation and absorption [8,66]. The concentrations of Ce^3+^ of all the NiCex-C catalysts were calculated by adding the areas under each deconvolution peak and dividing by the total peak area. The detailed molar ratio of Ce^3+^/(Ce^3+^ + Ce^4+^) was tabulated in Table 4. It could be found the highest Ce^3+^ concentration of 23.2% for the NiCe5-C catalyst was obviously higher than other Ce-containing catalysts, demonstrating that there were more oxygen vacancies at the Ni–O-Ce interfaces. In a word, the NiCe5-C catalyst owned the most active sites for CO_2_ absorption and activation, as well as surface Ni^0^ sites for the H_2_ molecular splitting. They may work synergistically and efficiently in CO_2_ hydrogenation under this reaction condition. 

### 3.5. Catalytic Activity and Stability in CO_2_ Methanation Reaction 

The catalytic performance of all prepared NiCex-C (x = 0, 1, 5, 10) catalysts for the CO_2_ methanation reaction were investigated at gas hourly space velocity (GHSV) of inlet gas of 15,000 mL/g_cat_/h with an H_2_/CO_2_ molar ratio of 4.0 in the temperature range of 200–350 °C. It was known that the CO_2_ methanation process was exothermic, so the total conversion could not be reached to 100% at high reaction temperature (>200 °C) [37]. CO_2_ methanation performances of the NiCex-C catalysts were shown in Figure 9. As shown in Figure 9a, CO_2_ conversion increased for all the samples with increasing reaction temperature. A significant difference in performance was displayed at 250 °C. CO_2_ conversion increased at 250 °C from 11.7% for NiCe0-C to 30.2% and 80.0% for NiCe1-C and NiCe5-C, respectively. The maximum CO_2_ conversion was observed at 300 °C. Then the catalysts reached CO_2_ equilibrium conversion above 300 °C. T_50_ was used to identify the low-temperature activity of the samples [11]. The value of T_50_ was 263.8 °C and 258.4 °C for the Ce-free NiCe0-C catalyst and the NiCe1-C catalyst, respectively, while it was obviously decreased to 238.6 °C for theNiCe5-C catalyst, indicating the promotion effect from Ce. However, when further increasing Ce content, the value of T_50_ increased to 277.9 °C for the NiCe10-C catalyst. It could be found that the catalytic performance first increased and then decreased with the increase of Ce content. This excellent low-temperature performance of NiCe5-C was from two main aspects: (I) Abundant active Ni^0^ sites evenly dispersed over the NiMgAl mixed metal oxide surface (XPS and TEM results), which provided active metal sites for H_2_ molecular dissociation and further promoted the hydrogenation process [41]. (II) The medium basic sites were conducive to the CO_2_ adsorption and activation [20]. NiCe5-C possessed the highest amounts of medium-strength basic sites (CO_2_-TPD result). Nevertheless, NiCe10-C exhibited the worst hydrogenation activity. This could have been mainly caused by the lowest amounts of total basic sites, as confirmed by CO_2_-TPD. Furthermore, CO as the main byproduct in the CO_2_ methanation process was due to the reverse water gas shift (RWGS) process: CO_2_ + H_2_ = CO + H_2_O. It is an endothermic reaction favorable at high temperature [67]. The CH_4_ selectivity of NiCex-C catalysts presented a high value of nearly 100% (Figure 9b), and few CO byproducts could be found at low reaction temperatures. The detailed product distributions of the tested catalysts in the CO_2_ methanation reaction at a reaction temperature of 225 °C were displayed in Table 5. At 225 °C, the CO_2_ conversion of the Ce-free sample was only 3.4%, while that of NiCe5-C reached a maximum value of 13.9%. The CO_2_ conversion rate (*R*_CO2_) and the TOF (turnover frequency) values of NiCe5-C were 4.1 times and 3 times higher than that of NiCe0-C, respectively. It indicated that the low-temperature activity of the Ni-based catalysts derived from hydrotalcite-like precursors could be significantly enhanced by moderate Ce doing. The activity of Ni-based catalysts for low-temperature CO_2_ methanation reported in the literature over the last decade was also compared, as shown in Table 6. The CO_2_ conversion of the NiCe5-C catalyst in this work is better than those reported in other literature [13,34,68,69,70,71,72,73] at a low temperature of 250 °C.

The long-term stability of the catalyst was of great importance for industrial application. Catalysts are prone to sintering under high-temperature conditions, which can lead to catalyst deactivation [3]. Therefore, the stability of the catalyst was judged by testing the CO_2_ conversion and CH_4_ selectivity at higher reaction temperatures (300 °C) and whether there were significant changes under prolonged operation as indicators. After 80 h of the stability test, as shown in Figure 10, the NiCe5-C catalyst also exhibited high activity, with about 91.7% of CO_2_ conversion and almost 100% of CH_4_ selectivity. In addition, there was no obvious deviation for them under the 80-h stability test. It indicated that the well-dispersed nickel particles on the catalyst surface (TEM results) could enhance the anti-sintering ability of the NiCe5-C sample and avoid deactivation after long-term use. The XRD spectra of the NiCe5-C catalyst after the 80-h stability test was shown in Figure 11. The crystalline phase remained essentially the same compared with that of the fresh NiCe5-C catalyst, which proves the stability of the catalyst structure after this reaction. 

Hydrogenation of CO_2_ to methane was a first order reaction, and the reaction of H with CO_2_ was the rate-controlling step for this reaction [74]. The apparent activation energies of the samples were measured by the Arrhenius equation. As shown in Figure 12, the logarithm of CO_2_ conversion as x variable and the reciprocal of temperature as y variable fit a straight line. The apparent activation energies of the catalysts were obtained by calculating the slope of the fitting lines. Compared to the NiCe0-C catalyst (98.13 kJ/mol), the apparent activation energy of the NiCe5-C catalyst was lower (96.04 kJ/mol). This result suggested that the CO_2_ methanation reaction was more facile on the NiCe5-C catalyst. It was contributed to the remarkable promotion effect of cerium on promoting charge transfer from active metal to the CO_2_ molecules [73]. Therefore, the reactant CO_2_ molecules were more easily activated, which is in favor of decreasing the energy barrier of the H and CO_2_ reaction.

### 3.6. The Descriptors of the Relationship between Catalytic Performance and Surface Basicity 

On the basis of previous literature reports, the Lewis acid-base sites instead of strong basic sites were involved in the CO_2_ methanation mechanism [75,76]. Therefore, it was necessary to explore the relationship between catalytic activity and the number of medium basic sites. As plotted in Figure 13a, the highest CO_2_ conversion at 225 °C was achieved over the NiCe5-C with the strongest medium-strength basic sites. In addition, the CO_2_ conversion of the catalysts was positively correlated with the number of medium basic sites. In this contribution, we observed the turnover frequency of the catalysts almost displayed linear relationship with the content of medium-strength basic sites at 225 °C (Figure 13b). It showed that the TOF values increased with increasing amounts of medium-strength basic sites. These results demonstrate that the variation of Ce content could effectively regulate the number of medium-strength basic sites on the surface of NiCex-C, and the medium-strength basic sites on the catalyst played a crucial role in achieving high catalytic activity for CO_2_ methanation at low temperatures.

## 4. Conclusions

In this work, we demonstrated an extremely straightforward method of preparing Ce-promoted NiMgAl hydrotalcite-derived catalysts via a co-precipitation operation to promote CO_2_ methanation at low temperatures. The NiCe5-C sample exhibited superior low-temperature activity for CO_2_ methanation (CO_2_ conversion: 80.0%, CH_4_ selectivity >99%, GHSV = 15,000 mL/gcat/h, H_2_/CO_2_ = 4, 250 °C, 0.1 Mpa), together with high stability, while the CO_2_ conversion of the Ce-free sample was only 11.7% under the same conditions. At 225 °C, the TOF value of NiCe5-C (11.9 h^−1^) was three times higher than that of NiCe0-C (3.9 h^−1^). The characterization results indicated that the incorporation of the appropriate amount of Ce into NiMgAl could weaken the strong metal-support interaction, thereby promoting the reduction of the NiO species. Meanwhile, the Ce element could efficiently regulate the surface basic species. The medium-basic sites were found to have a good linear correlation with TOF. The NiCe5-C catalyst with the highest Ni^0^/(Ni^2+^ + Ni^0^) ratio and abundant medium-basic sites was found to be exceptionally effective at dissociating H_2_ and activating CO_2_. In addition, it was able to hydrogenate methane with great efficiency. Our contribution provided a promising and practical thought to raise the catalytic performance in CO_2_ methanation at low temperature.

## Figures and Tables

**Figure 1 materials-16-04642-f001:**
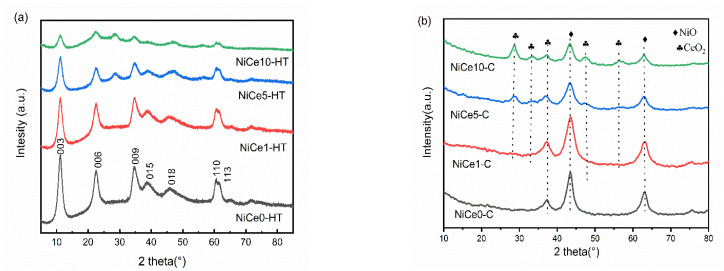
XRD patterns of the hydrotalcite-like precursors NiCex-HT (**a**) and the mixed oxides NiCex-C (**b**).

**Figure 2 materials-16-04642-f002:**
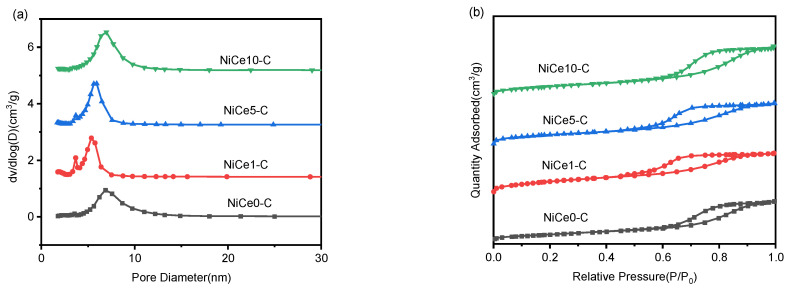
N_2_ adsorption-desorption isotherms (**a**) and distributions of the pore size of the NiCex-C catalysts (**b**).

**Figure 3 materials-16-04642-f003:**
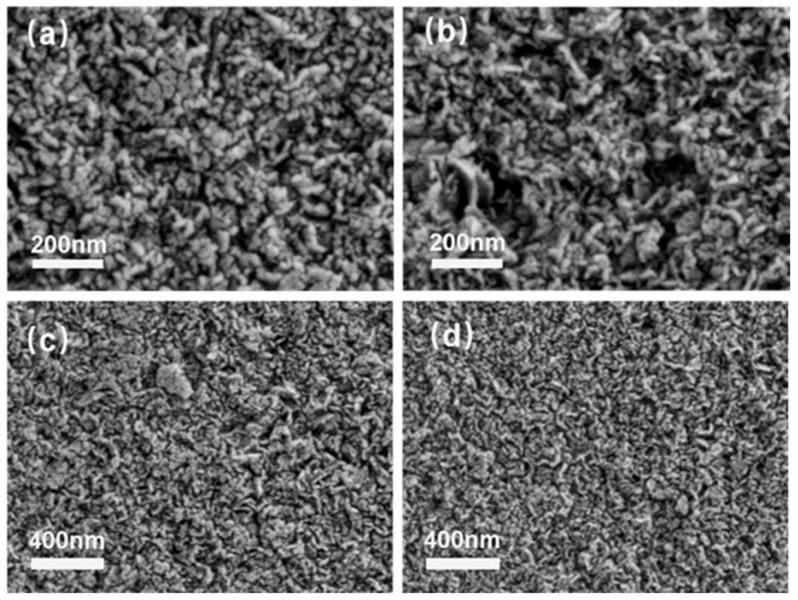
SEM images of (**a**,**c**) NiCe0-HT, (**b**,**d**) NiCe5-HT.

**Figure 4 materials-16-04642-f004:**
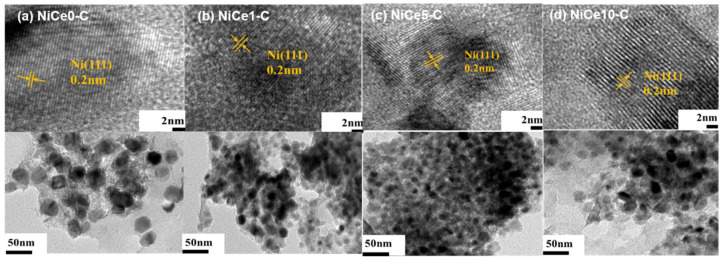
TEM and HRTEM images of the catalysts (**a**) NiCe0-C, (**b**) NiCe1-C, (**c**) NiCe5-C, and (**d**) NiCe10-C.

**Figure 5 materials-16-04642-f005:**
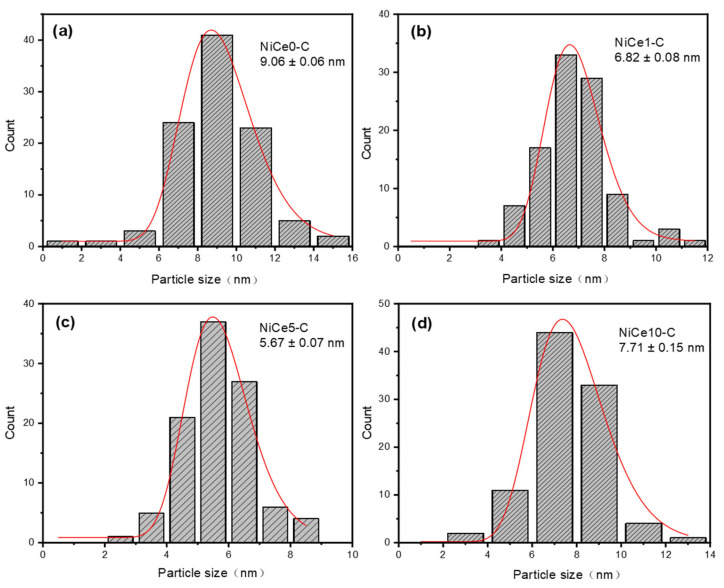
Metal particle size distribution over the reduced catalysts by TEM images, (**a**) NiCe0-C, (**b**) NiCe1-C, (**c**) NiCe5-C, and (**d**) NiCe10-C.

**Figure 6 materials-16-04642-f006:**
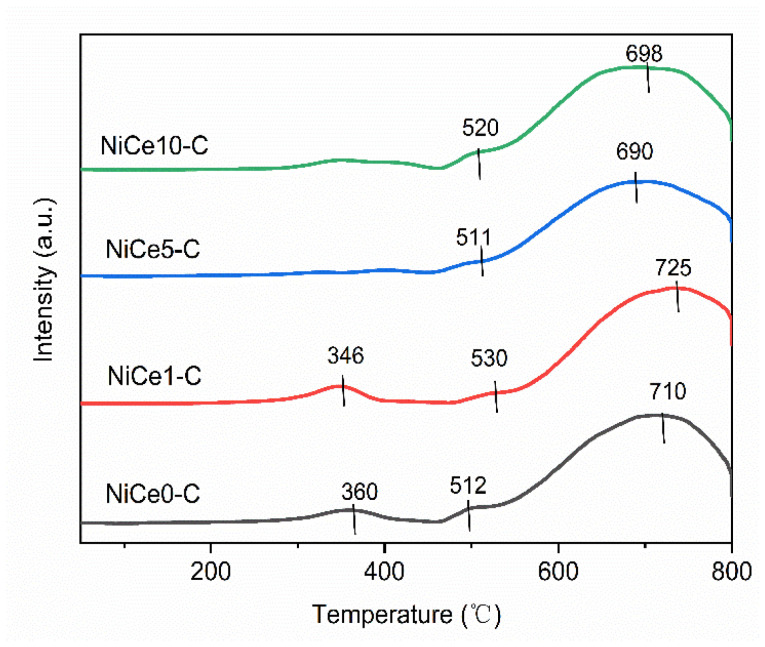
H_2_-TPR profiles of NiCex-C (x = 0, 1, 5, 10) catalysts.

**Figure 7 materials-16-04642-f007:**
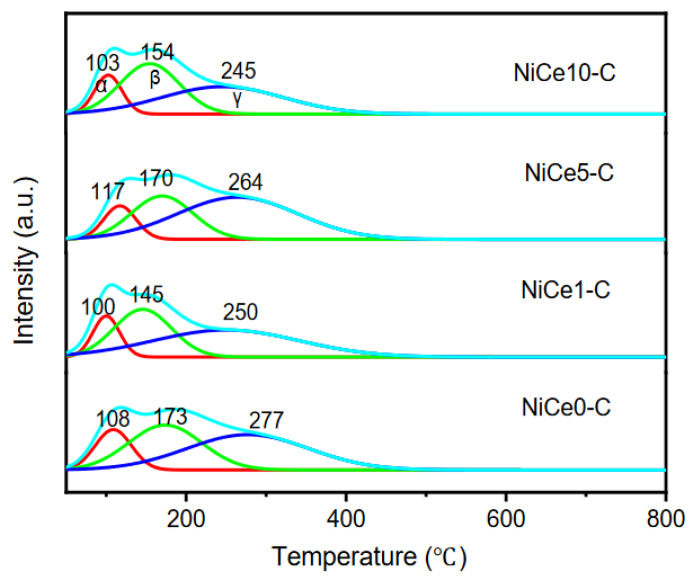
CO_2_-TPD profiles of the NiCex-C catalysts; the red line indicated the α pick, the green line indicates the β-peak and the blue line indicates the γ-peak.

**Figure 8 materials-16-04642-f008:**
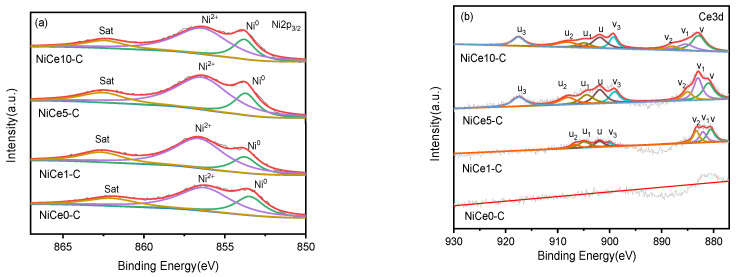
XPS spectrum of Ni 2p_3/2_ (**a**) and Ce3d (**b**) of the NiCex-C catalysts. In (**a**) the green line indicates the Ni^0^ species, the purple line indicates the Ni^2+^ species, the brown line indicates satellite peaks, and the red line is the original curve.

**Figure 9 materials-16-04642-f009:**
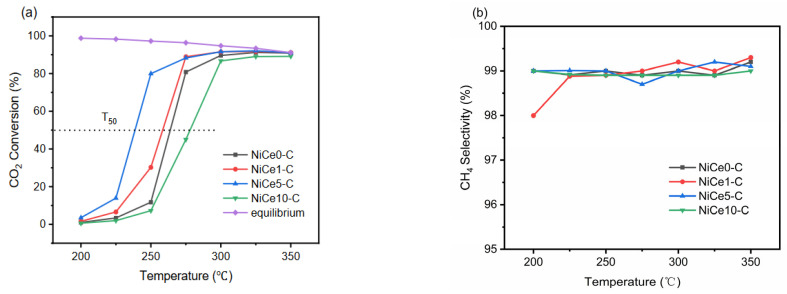
CO_2_ conversion (**a**,**b**) CH_4_ selectivity of NiCex-C catalysts with a temperature range from 200 to 350 °C, GHSV = 15,000 mL/g_cat_/h H_2_/CO_2_ = 4 (molar ratio), 50 mL/min, 200 mg catalyst.

**Figure 10 materials-16-04642-f010:**
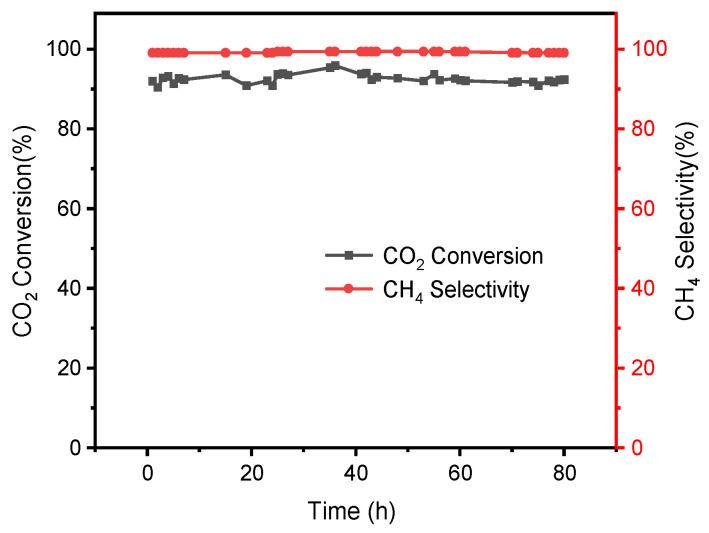
Stability for CO_2_ methanation over the NiCe5-C catalyst at 300 °C. Reaction conditions: H_2_/CO_2_ = 4:1, GHSV = 15,000 mL/g_cat_/h, 0.1Mpa.

**Figure 11 materials-16-04642-f011:**
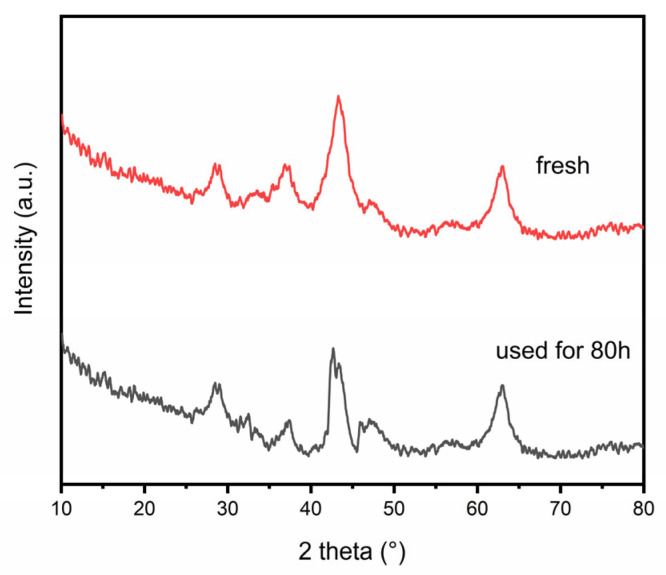
Comparison of XRD patterns of the NiCe5-C catalyst before and after the 80-h stability test.

**Figure 12 materials-16-04642-f012:**
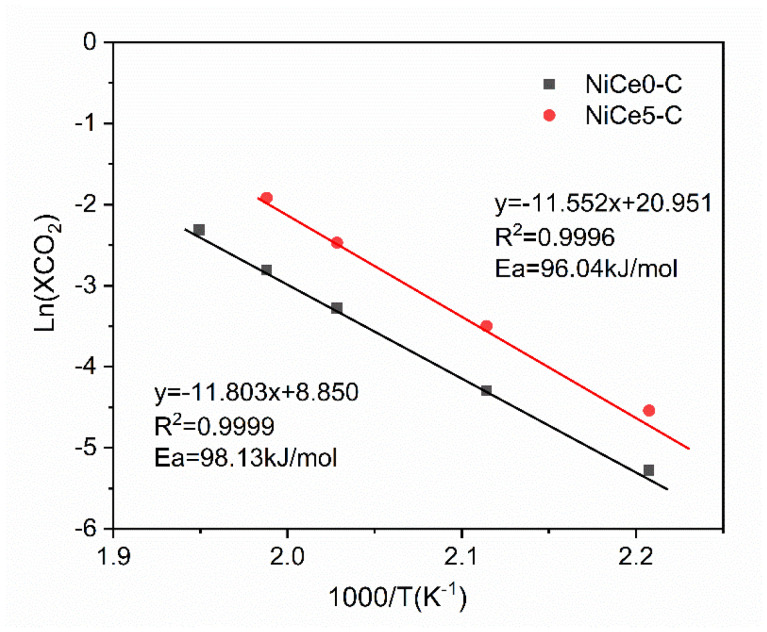
Arrhenius plots for the NiCe0-C and NiCe5-C catalysts.

**Figure 13 materials-16-04642-f013:**
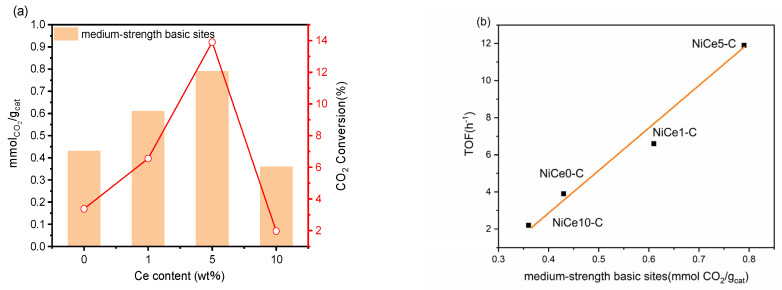
(**a**) Effect of Ce content on the medium-strength basic sites and CO_2_ conversion (T = 225 °C). (**b**) The relationship between medium-strength basic sites and TOF (T = 225 °C).

**Table 1 materials-16-04642-t001:** Detailed information about the textural properties of the NiCex-C catalysts.

Sample	S_BET_ ^a^ (m^2^/g)	V_p_ ^b^ (cm^3^/g)	D_p_ ^c^ (nm)	Ni ^d^ %	Ce ^e^ %	Mg/Al ^f^
NiCe0-C	127.7	0.24	6.41	44.5	0.0	1.5
NiCe1-C	215.0	0.25	6.25	40.5	0.79	1.5
NiCe5-C	176.9	0.26	5.24	40.1	4.07	1.5
NiCe10-C	169.0	0.29	4.37	42.3	8.70	1.4

^a^ BET specific surface area. ^b^ Calculated from the adsorption amount of N_2_ at a relative pressure (P/P0) of 0.98. ^c^ BJH Desorption average pore diameter. ^d–f^ Determined by ICP-OES measurement. Actual Ni, Ce content, and Mg/Al.

**Table 2 materials-16-04642-t002:** Fitting analysis and parameters of CO_2_-TPD for all reduced NiCex-C samples.

Samples	Reduction Temperature (°C)			Relative Content (%)		
	α	β	γ	α	β	γ
NiCe0-C	108	173	277	16.1	36.2	47.7
NiCe1-C	100	145	250	13.9	35.5	50.6
NiCe5-C	117	170	264	11.9	30.3	57.8
NiCe10-C	103	154	245	14.3	40.5	45.2

**Table 3 materials-16-04642-t003:** Surface basic sites’ density results of CO_2_-TPD for all reduced NiCex-C samples.

Samples	Weak-Strength Basic Sites (α + β) (mmol CO_2_/g_cat_)	Medium-Strength Basic Sites (γ) (mmol CO_2_/g_cat_)	CO_2_-Adsorption Amount (mmol CO_2_/g_cat_)
NiCe0-C	0.48	0.43	0.91
NiCe1-C	0.59	0.61	1.20
NiCe5-C	0.58	0.79	1.37
NiCe10-C	0.44	0.36	0.80

**Table 4 materials-16-04642-t004:** XPS result of Ni^0^ and Ce^3+^ relative content for the NiCex-C catalysts.

Samples	Relative Content (%)	
	Ni^0^/(Ni^0^ + Ni^2+^)	Ce^3+^/(Ce^3+^ + Ce^4+^)
NiCe0-C	31.5	0.0
NiCe1-C	34.7	19.6
NiCe5-C	40.5	23.2
NiCe10-C	34.1	18.8

**Table 5 materials-16-04642-t005:** Comparison of the catalytic performance of NiCex-C (x = 0, 1, 5, 10) catalysts in.CO_2_ methanation at 225 °C.

Samples	Conversion (%)	Selectivity (%)		*R*CO_2_ (μmolCO_2_/g_cat_/s)	*TOF* (h^−1^)
		CH_4_	CO		
NiCe0-C	3.4	98.7	1.3	1.26	3.9
NiCe1-C	6.6	98.9	1.1	2.43	6.6
NiCe5-C	13.9	99.0	1.0	5.17	11.9
NiCe10-C	2.0	98.9	1.1	0.73	2.2

Reaction conditions: T = 225 °C (conversion < 15%), GHSV = 15,000 mL/g_cat_/h, H_2_/CO_2_ = 4 (molar ratio), 50 mL/min, 200 mg catalyst.

**Table 6 materials-16-04642-t006:** Comparison of catalyst activity for CO_2_ methanation reactions in the last decade.

Catalyst	Temperature (°C)	The Inert Gas Ratio of H_2_/CO_2_	CO_2_ Conversion (%)	CH_4_ Selectivity (%)	Reference
NiCe5-C	250	H_2_:CO_2_ = 4	80.0	99.0	This work
Ni20Fe1.5	250	H_2_:CO_2_ = 4	71.0	99.0	[68]
HTNi15Cu1	250	H_2_:CO_2_ = 4	9.0	89.0	[69]
Ni21La0.4	250	H_2_:CO_2_ = 4	55.0	99.0	[70]
Ni42.5	250	H_2_:CO_2_ = 4	72.0	99.9	[34]
Ni/CeO_2_	250	H_2_:CO_2_ = 4	30.0	99.9	[71]
Ni–La/SiC	250	H_2_:CO_2_ = 4	39.6	99.6	[72]
12Ni6Ce/CNT	250	H_2_:CO_2_ = 4	32.0	99.9	[73]
NiW1MgOX	250	H_2_:CO_2_ = 4	38.0	99.9	[13]

## Data Availability

No new data created.

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
