# Peer review of "Engineering the Quaternary Hydrotalcite-Derived Ce-Promoted Ni-Based Catalysts for Enhanced Low-Temperature CO2 Hydrogenation into Methane"

_materials, 2023, doi:10.3390/ma16134642_

Round 1

Reviewer 1 Report

The manuscript entitled “Engineering the quaternary hydrotalcite derived Ce-promote Ni-based catalysts for enhanced low-temperature CO2 hydrogenation into methane”. It is an interesting topic in academic research that works with heterogeneous catalysis. That said, the manuscript reports the synthesis of mixed oxide catalysts with extensive characterization. That said, the manuscript reports the synthesis of mixed oxide catalysts with extensive characterization. And that doping with a nominal 5% Ce imparts improved low-temperature catalytic performance in the hydrogenation of CO2 to methane. Therefore, I recommend the acceptance of the same in Journal Materials. However, I have a few small suggestions and comments before final acceptance:

Highlight the novelty of the work in relation to those already found in the literature

Line 92-93 ......,therefore the CO2 conversion was significantly accelerated at low reaction temperatures. Enter this temperature to improve the reading.

Because it is underlined, metal-support interaction??; sodium hydroxide solution ?? operated ??, Moreover????, smaller??, calcination??, calcination??  throughout the manuscript there are other underlines.

Correct ml to mL

In Table 1, it is evident that the real Ce% load are all below the nominal loads, comment on this observation in the manuscript. This will reinforce the Ce impregnation method in the HT structure.

The manuscript emphasizes its efficiency as a catalyst with Ce (5 wt%). But little was mentioned about these results compared to others already reported, such as: activation energy if it is in the same magnitude of other catalysts.

Comparison with literature data would greatly improve the quality of the work and readers' understanding of the catalyst's efficiency with those already reported.

Was there no catalyst stability study below 300 °C? As in the temperature at which it presented the best results.

The characterization of the catalyst after 80 h of operation would be important to corroborate its stability.

The manuscript needs a major review, as they have many words together with quotations, with units, underlined expressions.

Reviewer 2 Report

Dear authors,

This is a very hot topic, and the extensive investigation of the synthesized materials, would be of great interest in the next few years. The spectroscopic and other techniques used, were previously shown to be very informative, however, at this stage, i would recommend MAJOR revisions, as several mistakes exist, which should be corrected. See attached .pdf file, where for your convenience, I've included my comments. 

English language should be improved, in addition to many typo errors exist. 

Reviewer 3 Report

The manuscript has been reviewed. In general, it is a high quality manuscript with well designed experimental works in supporting the conclusions. There are just a few minor comments to help to improve the manuscript.

1)In Line 23, the acronym TOF should be defined when it was first appeared in the manuscript.

2)In line 199,the authors wrote as "the morphology, metal dispersion and the lattice spacing ", but there is no metal dispersion data reported from TEM in result. Also, the authors need to put some details on how to obtain the frequency distribution histogram of Ni-particle size, as given in Line 269

3)In line 214, the authors explained that the hydrotalcite structure was destroyed and obtained the mixed metal oxide, while it was mentioned in in line 233 that the pore structure might be caused by the stack of 2D structure of HT. These two claims seen in conflict.

4)Line 287 and 289, the authors mentioned the low temperature reduction peak at 350 oC, but Figure 6 gave as 360 oC.

5) Figure 9(b) can be improved. Since all the CH4 selectivity are close to 100%, the y scale can be change to for example between 95-100% so that a clear difference between each sample can be observed.

The English is fine.

Round 2

Reviewer 1 Report

The manuscript now looks much better than the first version. The current version reads well. I see that my suggestions were met for the most part. I suggest that the justifications described for my questions could all be in full in the manuscript. It will certainly improve the quality of the work even more, is a suggestion. That said, I congratulate the work and recommend its publication in Journal Materials.